# Benchmarking human visual search computational models in natural scenes: models comparison and reference datasets

**Fermin Travi**[1*]
fermintravi@gmail.com

**Gonzalo Ruarte**[1*]
gonzalorpg@gmail.com

**Gaston Bujia**[1]
gbujia@dc.uba.ar

**Juan E. Kamienkowski**[1,2]
juank@dc.uba.ar

[1] Laboratorio de Inteligencia Artificial Aplicada, Instituto de Ciencias de la Computación, Universidad de Buenos Aires - CONICET, Argentina

[2] Maestría de Explotación de Datos y Descubrimiento del Conocimiento, Universidad de Buenos Aires, Argentina

## Abstract

Visual search is an essential part of almost any everyday human goal-directed interaction with the environment. Nowadays, several algorithms are able to predict gaze positions during simple observation, but few models attempt to simulate human behavior during visual search in natural scenes. Furthermore, these models vary widely in their design and exhibit differences in the datasets and metrics with which they were evaluated. Thus, there is a need for a reference point, on which each model can be tested and from where potential improvements can be derived. In the present work, we select publicly available state-of-the-art visual search models in natural scenes and evaluate them on different datasets, employing the same metrics to estimate their efficiency and similarity with human subjects. In particular, we propose an improvement to the Ideal Bayesian Searcher through a combination with a neural network-based visual search model, enabling it to generalize to other datasets. The present work sheds light on the limitations of current models and how potential improvements can be accomplished by combining approaches. Moreover, it moves forward on providing a solution for the urgent need for benchmarking data and metrics to support the development of more general human visual search computational models.

## 1 Introduction

Nowadays, in the field of Neuroscience, the mechanisms behind the visual processing of an image in the center of the visual field with no eye movements have been thoroughly studied [1, 2]. At the same time, the field of Artificial Intelligence has developed precise models of image processing and feature extraction (both objective and subjective), including artificial neural networks. These were inspired by and, in many cases, resemble visual processing in the brain. Among their recent successes, they have been capable of providing highly effective models which can predict gaze positions at an image when it is being freely explored [3, 4, 5, 6].

---

*Indicates equal contribution.

3rd Workshop on Shared Visual Representations in Human and Machine Intelligence (SVRHM 2021) of the Neural Information Processing Systems (NeurIPS) conference, Virtual.

Nonetheless, there are two fundamental aspects where current computational models are yet to provide answers. Firstly, when a biological system (mainly humans) observes an image, it is generally done with a goal in mind or to fulfill a task, such as memorizing the image or searching for an object. Secondly, it is not only relevant where the eye fixates upon, but also in what order it scans the image: eye movements do not follow random behavior, but rather different strategies which intend to minimize the number of fixations necessary to reach a goal [7, 8, 9, 10].

Currently, there are diverse algorithms that attempt to predict the sequence of eye movements (called *scanpath*) when looking for an object in a natural scene [11, 12, 13, 14]. However, the theoretical frameworks behind these models vary widely from each other. For instance, there are models which employ convolutional or recurrent neural networks [12, 13], others utilize reinforcement learning [14], and a third category applies a Bayesian approach [11, 15, 16]. Likewise, the datasets in which these models were evaluated can be categorized by the difficulty of the task (measured by the number of fixations made to find the target), the content of the images (absence or presence of text, human faces, and other distractors), how the target is presented, and so on. These differences can also be found in the methodologies employed to evaluate the results obtained. All of this hinders the task of concluding what works best and what needs further study. Exploration models greatly benefited from the construction of a saliency benchmark [17, 18][2], and the field of visual search is in need of one.

In the present work, we aimed to compare state-of-the-art visual search models in natural scenes. We selected three publicly available visual search in natural scenes algorithms [11, 12, 14], alongside the datasets they were tested in, with the addition of another published dataset [13], and we define common metrics for their due comparison. Going further, we discuss their differences and make the necessary adjustments to bring them together into a single pipeline.

## 2 Methods

### 2.1 Datasets

Each dataset comprises a set of search images, target objects for each image, and human subjects' scanpaths on those images (Table S1).

First, there are some differences between them in regard to experiment design. The *Interiors* dataset (which corresponds to the IBS family models) uses templates as targets and observers have to look for a specific object [11]. Both *MCS* and *COCOSearch18* state the object category ("microwave" and "clock" for the first, "car", "bottle" and others for the latter) and participants have to look for a broad object category (which could be in the image or not) [12, 14], and in the *Unrestricted* dataset (the one used for the IVSN model in natural scenes) a generalized version of the object is presented, so participants have to look for an object in a narrower category defined by the template (for example, a shoe or a stroller) [12]. Second, not all of the human scanpaths finish just as soon as a fixation from the participant lands on the target. In *MCS*, *COCOSearch18* and the *Unrestricted* dataset, participants have to press a button to finish the trial, so they can fixate on the target many times before they decide it is there. In the *Interiors* dataset, the trial finishes when the participants fixate on the target or after a varying upper limit of saccades is reached. Third, the criteria of when the target was found also varies across datasets. When compared to their corresponding models, the authors of *MCS* and *COCOSearch18* decide that the target was found if the average position during a fixation lands inside the target's bounding box (with no regard of the fixational eye movements or the size of the fovea), while Sclar et al. reduce the fixations to a grid using cells whose size is estimated from the fovea, and Zhang et al. use a square window, centered on the current target, whose size corresponds to two times the mean width and height of all the targets' bounding boxes. Fourth, the content of the images vary across datasets. For instance, the *Unrestricted* dataset is completely unconstrained and the only one to include human figures (faces or bodies) in them, the *Interiors* dataset only includes images of crowded interiors which could have several objects similar to the target (distractors), and both *MCS* and *COCOSearch18* are the only ones with colored images, resulting in shorter scanpaths. And, fifth, there was a huge variability in the number of participants (10 to 57) and images (134 to 2489).

Thus, we carried out a preprocessing of these datasets in order to fit them to a common format, equalized them as much as possible, and set a common criteria for all of them (see Appendix A.2).

---

[2]https://saliency.tuebingen.ai/

## 2.2 Visual Search Models

We analyzed three models: the first one is the Invariant Visual Search Network (IVSN) [12][3]. IVSN is based on convolutional neural networks, due to their resemblance to humans' visual cortex, and performs zero-shot invariant visual search. It makes use of two different versions of a pretrained VGG16: one for the search image, which is used to extract bottom-up features (visual ventral cortex network), and a second one for the target image, which is used to extract top-down features (prefrontal cortex network). Once these extractions have been done, the feature representation of the target image is used to modulate the representation of the search image through a convolution, generating what Zhang et. al. called an attention map. A greedy search is then performed on this attention map, which means the whole image is processed at once (contrary to how human vision works), there is no accumulation of information across saccades, and the inhibition of return is forced.

The second model is the correlation-based Ideal Bayesian Searcher (cIBS) [11][4]. cIBS attempts to minimize the number of fixations needed to find the target. First, it uses a saliency map of the search image computed with DeepGaze II [19] as prior. Second, at each step, the next fixation is calculated as the one that maximizes the likelihood of finding the target, taking into account all the information gathered in previous steps [20]. The likelihood is calculated through a visibility map (estimated as a 2D Gaussian centered on the current fixation) and a target similarity map, which is computed with the search image and the target image via cross-correlation [11]. Given the opportunity to test this model in several different datasets, we evaluated different versions of it, modifying its target similarity map. In particular, we replaced cross-correlation with SSIM (sIBS) and a convolutional neural network (nnIBS) . The latter is based on the attention map computed by IVSN. By performing this substitution, the resulting model is able to capture object invariance while maintaining a good performance on template matching. This variation was used against the other models.

The last model evaluated is based on Inverse Reinforcement Learning (IRL)[14][5]. It was trained on *COCOSearch18*, performing categorical visual search, and it is only capable of searching for an object that belongs to one of those 18 categories. In a similar manner to cIBS, a Gaussian blur is applied to the search image to account for loss of visibility in the periphery. Both, the unaltered search image and the blurred image, are preprocessed with a neural network pretrained on COCO2017 which performs a panoptic segmentation (Detectron2 [21]) and this is used as the model's input. On top of this, a one-hot encoding of the target object category is concatenated with the input of each layer in the network. This way, the authors aim to capture both the image and the target object's context. The code for the preprocessing of the images was not available, so we reproduced it from the references in the publication (see Appendix A.7 for details).

All of the models described were ported to Python (whenever necessary) in order to engulf them in a common framework. Additionally, the necessary changes to run them on the previously established dataset's format were made (Appendix A.5). Therefore, each model is now able to receive any of the four preprocessed datasets as input and the output follows the same criteria as that of human subjects', allowing for a correct comparison.

## 2.3 Metrics

We are interested in how the computational models compare against humans in terms of both efficiency and scanpath similarity. To analyze efficiency, we measured the proportion of targets found for a given number of fixations and the area under this curve (AUC) is reported. To analyze similarity, we use Multi-Match (MM) [22]. In short, Multi-Match represents scanpaths as geometrical vectors in a two-dimensional space (any scanpath is build up of a vector sequence in which the vectors represent saccades, and the start and end position of saccade vectors represent fixations). Two such sequences (which can differ in length) are compared on the four dimensions vector shape, vector length (saccadic amplitude), vector position, and vector direction for a multidimensional similarity evaluation. The temporal dimension is excluded as we are not considering fixation durations.

For each model, its scanpaths were compared to each subject's scanpaths and the result was averaged across subjects for each image (hmMM). Also, a human ground truth was estimated by comparing human subjects' scanpaths to each other and averaging the results across subjects for each image

---

[3] https://github.com/kreimanlab/VisualSearchZeroShot
[4] https://github.com/gastonbujia/VisualSearch
[5] https://github.com/cvlab-stonybrook/ScanpathPrediction

(whMM). This was done for every scanpath with length greater than two and in which the target was found (Tables S3 and S4). Difficulty varies across images, and so does whMM. To investigate if this variability is also present in the models, whMM and hmMM were plotted in two axes (Fig. 1D-E, 2E-H) and their correlation was calculated. These plots provide an easy overview of how well the model has captured human behavior: the closer the points are to the diagonal, the more exchangeable the model's scanpaths are with those of human subjects, whereas the points further to the left represent those images in which within human similarity was greatest, but human-model similarity was not as high [11].

## 3   Results

Different sets of images, targets and tasks present different challenges to the models. These datasets all have crowded images, full of distractors (with different degrees of similarity), and with a strong context. Moreover, the task was slightly different (looking for an exact object or for a category) and, thus, the presentation of the target example that guides the search was different. The cIBS model was previously evaluated in looking for an exact object [11]. Briefly, this model consists of two parts: the prior that gives it a first gist of the image and an initial context, and the IBS rule to update the probabilities of finding the target, which is based on a similarity map that compares each image patch with the target. In order to allow the model to both generalize the target template and also to get confused/attracted by distractors (as humans are), we evaluated three different similarity maps: the correlation and the SSIM between each image patch and the target (cIBS and sIBS, respectively), and the attention map from the IVSN model (nnIBS). Performance was greatest with SSIM (Fig. 1A-C), albeit at a greater computational expense. Crucially, nnIBS was the only one capable of finding the generalized targets used in the *Unrestricted* dataset. This means it is able to perform categorical visual search and correctly generalize to the other datasets. Conversely, both cIBS and sIBS had to use cropped versions of the targets in the images (templates). Nonetheless, this modification did not alter scanpath similarity with human subjects (Table S3 and Fig. 1D-F).

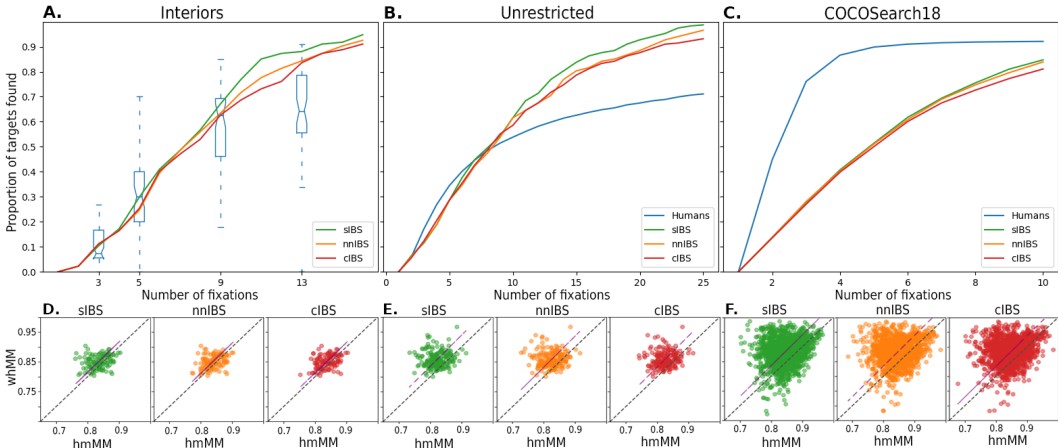

Figure 1: **A-C)** Ratio of targets found (vertical axis) for a given number of fixations (horizontal axis). In A), human performance is displayed as box plots due to the upper limits set on the number of saccades for human subjects (varying between 2, 4, 8 and 12). **D-F)** Mean MM values within humans (whMM) as function of the mean MM values between each human observer and a model (hmMM) (Table S3). Only trials where the target was found are considered.

Another modification that we performed to the models in order to adapt them to different datasets involved changing the patch size used for inhibition-of-return and target identification in IVSN. This model performs greedy search through an attention map built with neural networks. After an image patch is explored, the probabilities in that position are zeroed, forcing an inhibition-of-return effect [12]. Thus, it is strongly dependent on this patch size. We evaluated different values for it relative to the mean target size of each dataset, including the one proposed by the authors, and kept the best working version (Appendix B.2).

When comparing all of the visual search models, not surprisingly, each one performed best in its own dataset. As IRL depends on the categories of the *COCOSearch18* dataset, it could only be evaluated on a small portion of the *Unrestricted* and *Interiors* datasets. In both datasets, IRL's performance was particularly low, achieving under $50\%$ accuracy (19 of 46 and 17 of 42 successful trials in the *Interiors* and *Unrestricted* datasets, respectively). Additionally, even though it could run on the entire *MCS* dataset (for it only contains microwaves and clocks as targets), its performance plummeted in comparison to the other models and human subjects (Fig. 2D).

In all datasets, IVSN showed great search efficiency, mainly in the first few fixations, likely as a consequence of its wider view of the scene –i.e. the model doesn't include a decay due to the foveated structure of the eye–. This, alongside its independence between fixations (there is no accumulation of information), is probably why its similarity with human subjects never was the highest (Table S4).

Finally, nnIBS was the most consistent with human behavior in regard to scanpath similarity (Table S4), although it performed poorly in *COCOSearch18* (Fig. 2C). Remarkably, the distractors in the *Interiors* dataset seemed to be capable of fooling IVSN in some cases; that did not happen for nnIBS, even though it used the same attention map (Fig. S4). This could be due to its greedy behavior, which simply goes to the point that most closely resembles the target, regardless of distance or other factors (see also Fig. S3).

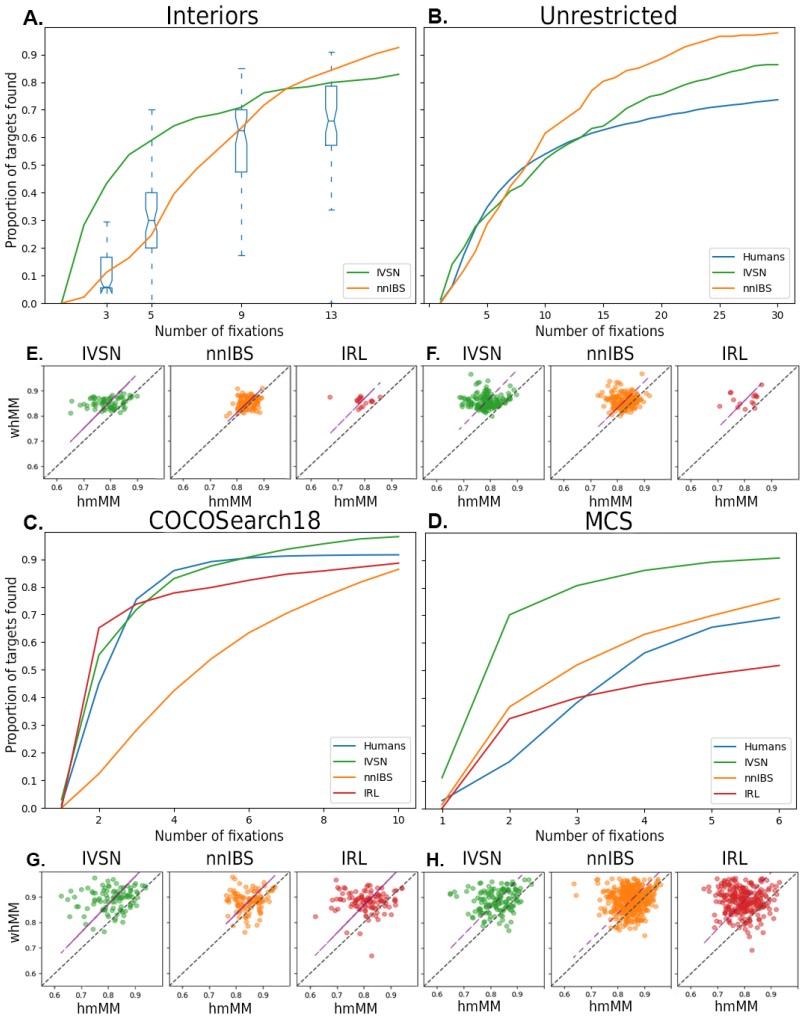

Figure 2: **A-D)** Cumulative performance curve for each model in each dataset. In A-B), IRL's curve is not plotted due to the low number of images tested. **E-H)** Mean MM values within humans (whMM) as function of the mean MM values between each human observer and a model (hmMM), for each dataset. Only trials where the target was found are considered.

In order to guide future improvements of visual search algorithms, we explored other trials where the models' scanpath dissimilarity with human subjects was greatest. On one hand, humans were able to rapidly understand the context of the image they were looking at and performed their search accordingly [23, 24, 25]. For instance, while looking for a car, they immediately understood it would be on the road and not in the sky or buildings (Fig. S5A, see also Fig. S5B-C for other examples). Meanwhile, none of the visual search models analyzed here were able to capture this behavior. On the other hand, it is also important to note that human subjects did not follow salient distractors (such as human faces while looking for something else), whereas nnIBS did, mainly in the first fixations (Fig. S6). This is because of its prior, which is a saliency map computed with DeepGaze II [19], a neural network trained on human subject data on free viewing.

## 4   Conclusions

Even though all of the models and datasets evaluated were designed on the task of visual search in natural scenes, the stark differences in the models' performance in different datasets show how much work still needs to be done in terms of providing a clear definition of the problem at hand and how to assess the different solutions. While categorical visual search provides the advantage of evaluating the ability to capture object invariance, searching for a specific target allows to evaluate the ability of identifying the target among several distractors belonging to the same category. The criteria used for data collection is also of relevance, where automatically stopping when fixating at the target may fail to provide an account of when human subjects actively recognize the target. Lastly, although colored images may be more realistic, they often result in shorter scanpaths than grayscale images (Table S1), due to the added information they possess.

With regard to the solutions themselves, our work shows some benefits and drawbacks of each paradigm considered. On the one hand, the IRL approach seems to be limited at present, as it is able to incorporate context information, but it does not generalize outside of its own dataset. On the other hand, while DNNs based models like IVSN appear to have better efficiency, the Bayesian approach produced the most human-like scanpaths. This is consistent with the idea of modeling human visual search as an active sampling process where successive steps attempt to reduce uncertainty about the localization of the target [26, 27]. Here, we suggest that combining these paradigms could potentially give rise to more precise and interpretable models. In particular, by modeling the visual system within a DNN framework [28], while applying a Bayesian approach for the central decision making [29, 30].

Notwithstanding, combining scene content with the target's meaning remained elusive for these state-of-the-art visual search computational models and proved to be particularly important for simulating human subjects' behavior. Further work should be carried out in order to develop general visual search algorithms capable of capturing context. nnIBS could benefit from modifying its prior by creating a map that correlates the target with the image's context. Recently, Levi and Ullman [31] have proposed an efficient manner of using relational reasoning to build context for small object detection, aggregating information across the entire image. This approach, named the Efficient Non Local module or ENL, could prove to be useful for this task.

The present work sheds light on both the limitations of current visual search algorithms and how potential improvements can be accomplished by combining approaches. Moreover, it is a first step into solving the urgency for the definition of a common set of metrics and data for the development of more general human visual search computational models.

## Author contributions

F.T. and G.R. prepared the datasets, coded the IBS models and adapted the other models, including numerical optimizations and speed-ups. F.T., G.R., G.B. and J.K. performed the analysis and wrote the manuscript.

## Acknowledgments

The authors were supported by the CONICET and the University of Buenos Aires (UBA). The research was supported by the CONICET (PIP 11220150100787CO), the ARL (Award W911NF1920240) and the National Agency of Promotion of Science and Technology (PICT 2018-2699).

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

# Appendix

## A  Methods and Materials

### A.1  Datasets description

Each dataset comprises a different set of search images, target objects for each image, and human subjects' scanpaths on those images (Table S1).

Table S1: Main characteristics of the datasets considered, highlighting the differences between them. Most datasets contain distractors of some sort, but, in this case, we are focusing on objects belonging to the same category as the target. Only scanpaths where the target was found are taken into account.

|  | **Interiors** | **Unrestricted** | **MCS** | **COCOSearch18** |
|---|---|---|---|---|
| #Subjects | 57 | 15 | 27 | 10 |
| #Images | 134 | 234 | 1790 | 2489 |
| #Scanpaths | $\sim$3.8K | $\sim$2.8K | $\sim$3.7K | $\sim$22K |
| Mean scanpath length | $5.2 \pm 2.36$ | $10.7 \pm 10.9$ | $3.76 \pm 2$ | $2.79 \pm 1.22$ |
| People | No | **Yes** | No | No |
| Exact target | **Yes** | No | No | No |
| Distractors | **Yes** | No | No | No |
| Color | No | No | **Yes** | **Yes** |
| Fovea size | 32x32 | 45x45 | 20x20 | 52x52 |
| Mean target size | 72x72 | 115x105 $\pm$ 68x71 | 73x82 $\pm$ 58x70 | 254x280 $\pm$ 126x139 |
| Image res. | 768x1024 | 1024x1280 | 508x564 | 1050x1680 |

### A.2  Equalizing datasets

To bring each dataset to a common format, a series of steps were carried out. To begin with, all of the human subjects' scanpaths were cut as soon as a fixation had landed on the target and only trials in which the target was found were considered. The same criteria was applied to the models. In order to do this, an estimation of the size of the fovea (i.e. the region around the fixation point in which the target could be identified) had to be done for each dataset (Table S1). With respect to the images, those in the *Unrestricted* dataset had to be transformed to grayscale, and the ones in the *MCS* dataset were rescaled to their average size. The targets' bounding boxes were not available for the latter, so they had to be retrieved from the COCO dataset's annotations, and target absent trials were excluded from the present analysis. Additionally, since only training trials of *COCOSearch18* are available, every image corresponding to testing was discarded, and a random subset was selected for evaluation to avoid overfitting.

Given the differences in the experiments' design, target images had to be cropped from the search images in both the *MCS* and *COCOSearch18* datasets, since two of the models evaluated need an image of the target to search. Conversely, images from the *Unrestricted* and *Interiors* datasets had to be categorized by hand as one of the COCO object categories depending on their target, so they could be processed by the visual search model trained on the *COCOSearch18* dataset. These differences were also present in the scanpaths' length of human subjects. Since every visual search model possesses a maximum number of fixations, this upper limit was set for each dataset according to the saturation of human performance.

Trivial search images (that is to say, those where the initial fixation lands on the target) were discarded in the *MCS* and *Unrestricted* datasets. In the case of the *Unrestricted* dataset, images were shown

twice to the participants, and here only the first trials were kept. *COCOSearch18* used some of the images for more than one trial, changing the target; these are regarded as different images.

Finally, a common format for human subjects' scanpaths and the input images and targets was defined, based on JSON files. The models' output follow the same format as those of human subjects. This way, a concordance in terms of nomenclature across all datasets and models is also achieved.

### A.3 Models' selection criteria

Additional visual search models were considered, such as the ones described in [13], [15], and [16], but they were discarded for either their code not being available or not admitting natural scenes images as input. However, we do intend to facilitate the incorporation of new models by publishing all of the code used here.

### A.4 Models' scanpath parameters

Despite the number of differences between models, we attempt to set a common criteria for deciding when the target was found. All of them use an oracle to decide if the target was found, and the search is carried out until the target is found or a maximum number of fixations is reached. It is worth noticing that it is not possible to apply precisely the same rule to every one of them, since IVSN runs over the original image and the IBS family and IRL run over different grids (Table S2).

Table S2: Description of how each model regards that the target was found.

| Model | Target Bounding Box | Fovea | End of scanpath criteria |
|-------|---------------------|-------|--------------------------|
| IBS | Square centered in the target. Projected to grid. | 1x1 in 24x32 grid | Fovea lands in bounding box |
| IVSN | Surrounds the target in original image size. Rectangular. | Mean target size in original image size | Fovea lands in bounding box |
| IRL | Surrounds the target in 320x512. Rectangular. | 1x1 in 20x32 grid | Fovea lands in bounding box (rescaled to 320x512) |

### A.5 IBS Python Implementation

The original code, written in MATLAB, was ported to Python, an open-source programming language available to everyone. From the start, the goal in mind was to be as clear as possible when it came to writing the code, including a precise definition of the input and output of each method. Additionally, we focused on an object-oriented approach, creating classes for each important concept within the model (prior, visibility map, target similarity map, and so on). All of this allows for different hypotheses to be tested with ease, including the resulting models in the present work, and future improvements (such as modifying the prior, in light of the results displayed here) can be implemented without much trouble. We believe well-written code is central for reproducible research.

On the other hand, since the model's hyper-parameters are tuned for images whose resolution is 768x1024, its input is automatically resized to this size, where the visibility and target similarity maps are computed, before downscaling it to a grid.

### A.5.1 Prior estimation

The estimation of the prior is done by computing saliency maps using the center bias over the input images, which in this case is done via DeepGaze II [19]. In order to run the model on several different datasets, the estimation of the prior is now incorporated in the model through a TensorFlow adaptation of a Jupyter Notebook provided by the BETHGE LAB[6].

---

[6]https://deepgaze.bethgelab.org/

### A.5.2 Cross-correlation in colored images

Given that the input of the original model was grayscale images, an adaptation had to be done in order to run the model on datasets where images are colored. Cross-correlation returns a different value for each color channel, so a weighted-average was calculated from common RGB to grayscale formulas. In particular, we chose the one used by the `scikit-image` library:

$$Y = 0.2125 * R + 0.7154 * G + 0.0721 * B$$

### A.5.3 SSIM

Due to the extensive use of SSIM in the video industry (to quantify for image quality, for instance), it arose as a natural replacement for cross-correlation when it came to computing the similarity between the search image and target. Moreover, it is capable of working in colored images.

SSIM is computed on the target and each possible region of the image. Then, these results are averaged and stored in the center pixel of the corresponding region. With this approach, black bars would remain at the edges of the images, so they are managed differently: we trim from the target those pixels that would overflow the image's bounds and then we compute SSIM between the target's leftovers and the corresponding region of the image. Since these calculations are made pixel by pixel, the computation behind this method is very expensive, but they have to be done only once (for the result of each target and search image is stored and can be reutilized).

### A.5.4 IVSN

By making use of our PyTorch implementation of the IVSN model, it became possible to adapt the attention map computed by IVSN[12]. Since its values range from zero to one, this adaptation was pretty straightforward: it simply serves as a replacement for cross-correlation. The advantages of capturing object invariance while retaining a good performance in searching for templates with little cost in computation made it our best choice for this benchmark.

### A.6 IVSN Python Implementation

The code available[7] is written both in MATLAB and LUA, using the first for pre and postprocessing and the latter for computations with the Caffe VGG16 model. Besides making the necessary adjustments to read the input and write the output in the previously defined JSON format, our Python implementation uses the PyTorch library alongside its implementation of VGG16.

Given the greedy nature of the algorithm, changes on the scanpaths produced were observed depending on small factors, such as using a different formula for the grayscale conversion of the input images or a different interpolation method for upscaling the neural network's output. Nonetheless, its performance remained unaffected (Fig. S1).

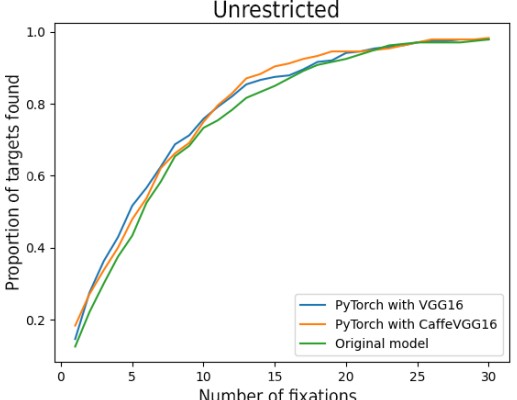

Figure S1: Performance comparison between the different versions of IVSN, in its own dataset, with the original target found criteria defined by the authors.

---

[7] `https://github.com/kreimanlab/VisualSearchZeroShot`

## A.7 Adapting the IRL model

This model is already written in Python[8], but its execution is not straightforward, since only the code for training the neural network is available. Moreover, the preprocessing of the search images described in [14] is also missing.

Following their description, in our implementation, input images are resized to 320x512 and a gaussian blur with $\sigma = 2$ is applied to them, producing two images for each search image. Both of these are fed forward through the Panoptic-FPN in Detectron2 [21] with Resnet-50 3X as backbone[9] and the output is resized to 20x32. We found differences in IRL's performance depending on how this downscaling was done. Detectron2 itself provides an output size as a parameter, but the performance was meager when compared to downscaling using common image interpolation techniques (such as nearest-neighbor), so the latter was used in our implementation. In all cases, we adopt the criteria of using the best performing model when some specifications are missing.

Once this has been done, masks are created from these semantic segmentations for each of the 134 COCO categories (if a category was not present in the search image, then its corresponding mask is simply the null matrix). These masks are the model's input.

All of this preprocessing has been integrated to the model itself, so our implementation is able to create the so-called Dynamic Context Belief maps on any given image. However, the categorization of the target object has to be done by hand.

# B Results

## B.1 Comparison of different similarity maps within the IBS family

Results for the different methods used in the target similarity map computation (Table S3): cross-correlation (cIBS), SSIM (sIBS), and neural network-based (nnIBS).

Table S3: IBS performance (AUC) and human similarity (MM) metrics as defined in 2.3. MM values are presented as the average (± standard error).

|  | AUC | Corr | AvgMM | MMvec | MMdir | MMlen | MMpos |
|---|---|---|---|---|---|---|---|
| **Interiors** | | | | | | | |
| nnIBS | 0.53 | **0.28** | **0.84** | **0.91**±0.02 | **0.71**±0.05 | **0.90**±0.03 | **0.83**±0.04 |
| sIBS | 0.56 | 0.12 | 0.83 | 0.90±0.02 | **0.71**±0.06 | 0.89±0.03 | **0.83**±0.04 |
| cIBS | **0.52** | 0.24 | **0.84** | **0.91**±0.02 | **0.71**±0.06 | 0.89±0.03 | **0.83**±0.04 |
| Humans | 0.42 | - | 0.85 | 0.92±0.01 | 0.73±0.05 | 0.91±0.02 | 0.84±0.03 |
| **Unrestricted** | | | | | | | |
| nnIBS | 0.63 | 0.017 | **0.82** | 0.90±0.02 | **0.67**±0.07 | 0.89±0.04 | **0.82**±0.04 |
| sIBS | 0.65 | 0.09 | **0.82** | 0.90±0.02 | **0.67**±0.07 | 0.89±0.04 | **0.82**±0.04 |
| cIBS | **0.62** | **0.10** | **0.82** | **0.91**±0.02 | **0.67**±0.06 | **0.90**±0.03 | **0.82**±0.04 |
| Humans | 0.56 | - | 0.86 | 0.93±0.01 | 0.72±0.06 | 0.93±0.02 | 0.84±0.05 |
| **COCOSearch18** | | | | | | | |
| nnIBS | **0.51** | 0.14 | **0.84** | **0.91**±0.02 | **0.68**±0.11 | 0.90±0.04 | **0.87**±0.04 |
| sIBS | **0.51** | 0.16 | **0.84** | **0.91**±0.02 | **0.68**±0.11 | 0.90±0.04 | **0.87**±0.05 |
| cIBS | 0.50 | **0.16** | **0.84** | **0.91**±0.02 | **0.68**±0.11 | **0.91**±0.04 | **0.87**±0.04 |
| Humans | 0.78 | - | 0.89 | 0.94±0.03 | 0.79±0.10 | 0.92±0.03 | 0.91±0.04 |

---

[8] https://github.com/cvlab-stonybrook/Scanpath_Prediction
[9] https://github.com/facebookresearch/detectron2

## B.2 Comparison of different patch sizes for the IVSN model

The behavior of IVSN is deeply ingrained with the patch size it uses for target identification and to apply the inhibition-of-return effect while searching in its attention map (Fig. S2).

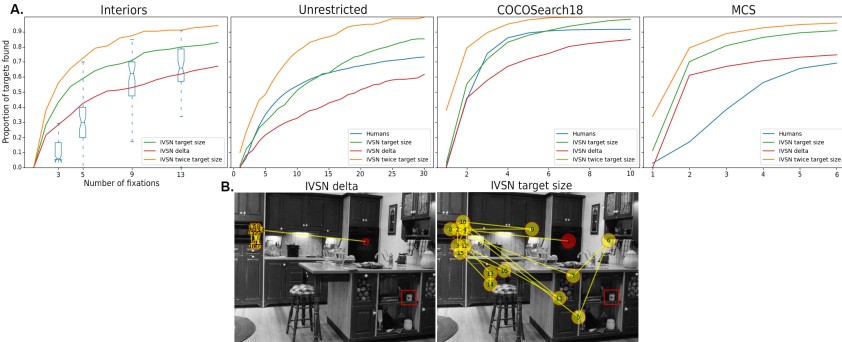

Figure S2: **A)** Three different patch sizes are tested for IVSN: the size of the human fovea (*delta*, in red), the mean target size (*target size*, in green) or twice the mean target size (original criteria used by Zhang et. al., in orange). **B)** A small patch size (left) causes the greedy algorithm to get stuck.

## B.3 Comparison between modeling approaches

Table S4: Models' performance (AUC) and human similarity metrics (MM) as defined in 2.3. MM values are presented as the average ($\pm$ standard error).

|  | AUC | Corr | AvgMM | MMvec | MMdir | MMlen | MMpos |
|---|---|---|---|---|---|---|---|
| **Interiors** | | | | | | | |
| nnIBS | 0.53 | **0.28** | **0.84** | **0.91**±0.02 | **0.71**±0.05 | **0.90**±0.03 | **0.83**±0.04 |
| IVSN | **0.65** | 0.23 | 0.79 | 0.87±0.05 | 0.68±0.08 | 0.82±0.10 | 0.78±0.06 |
| IRL | - | -0.08 | 0.79 | 0.88±0.03 | 0.65±0.06 | 0.85±0.05 | 0.79±0.05 |
| Humans | 0.42 | - | 0.85 | 0.92±0.01 | 0.73±0.02 | 0.91±0.02 | 0.84±0.03 |
| **Unrestricted** | | | | | | | |
| nnIBS | 0.63 | 0.017 | **0.82** | **0.90**±0.02 | **0.67**±0.07 | **0.89**±0.04 | **0.82**±0.04 |
| IVSN | **0.60** | 0.047 | 0.78 | 0.88±0.03 | 0.64±0.07 | 0.85±0.06 | 0.78±0.06 |
| IRL | - | **0.25** | 0.80 | 0.88±0.04 | **0.67**±0.05 | 0.85±0.07 | 0.79±0.05 |
| Humans | 0.56 | - | 0.86 | 0.93±0.01 | 0.72±0.06 | 0.93±0.02 | 0.84±0.05 |
| **COCOSearch18** | | | | | | | |
| nnIBS | 0.51 | 0.12 | **0.85** | **0.91**±0.02 | **0.69**±0.11 | **0.90**±0.04 | **0.88**±0.04 |
| IVSN | 0.80 | **0.19** | 0.81 | 0.88±0.05 | 0.67±0.13 | 0.86±0.08 | 0.84±0.07 |
| IRL | **0.75** | 0.02 | 0.81 | 0.89±0.04 | 0.64±0.14 | 0.87±0.07 | 0.83±0.08 |
| Humans | 0.78 | - | 0.89 | 0.94±0.03 | 0.79±0.10 | 0.92±0.03 | 0.91±0.04 |
| **MCS** | | | | | | | |
| nnIBS | 0.52 | 0.08 | **0.85** | **0.92**±0.03 | **0.70**±0.11 | **0.89**±0.05 | **0.88**±0.05 |
| IVSN | **0.75** | **0.18** | 0.81 | 0.89±0.04 | 0.67±0.12 | 0.85±0.08 | 0.84±0.07 |
| IRL | 0.38 | -0.06 | 0.79 | 0.88±0.04 | 0.64±0.12 | 0.84±0.07 | 0.81±0.07 |
| Humans | 0.42 | - | 0.89 | 0.95±0.02 | 0.75±0.11 | 0.93±0.04 | 0.92±0.04 |

## B.4 Comparison between modeling approaches: some failure examples

Here, we present examples which illustrate some failures of the models. In every figure, initial fixations are displayed in red, while participants' scanpaths are superimposed with different colors.

Figure S3 depicts a case where there is a sheer difference between IVSN and human subjects in terms of saccade amplitude, highlighting the relevance of modeling visual decay in the periphery.

In Figure S4, two different examples are shown where IVSN fell for distractors similar to the target, whereas nnIBS proved to be more robust.

Finally, Figure S5 shows how human subjects are capable of incorporating both the context of the image and the target's meaning to guide their search accordingly, a behavior that remains elusive for current visual search models.

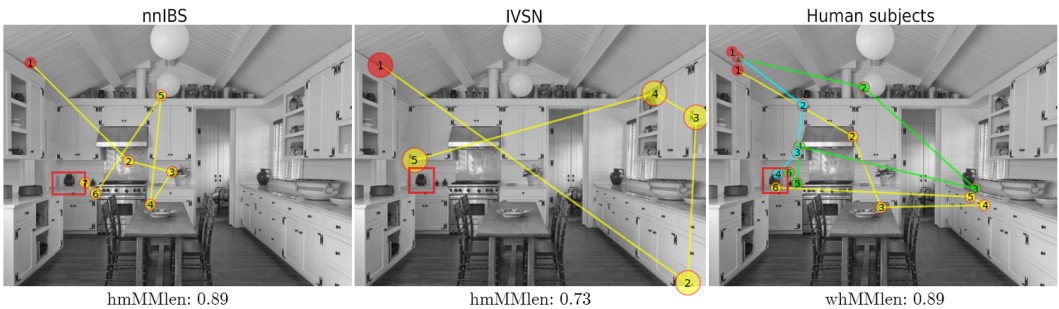

Figure S3: IVSN, due to its lack of a model of the fovea, crosses the entire image in one saccade.

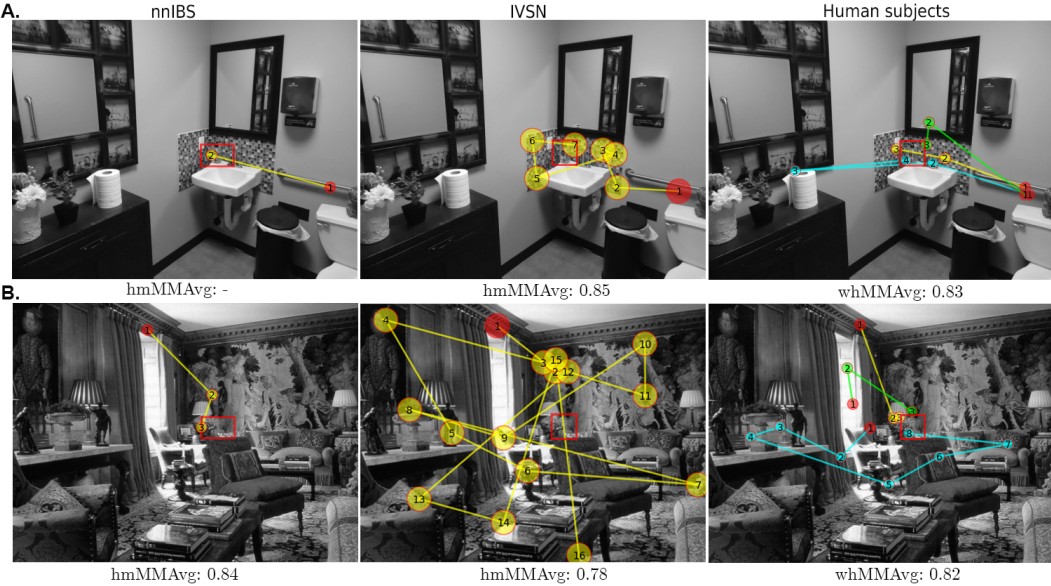

Figure S4: Even though nnIBS and IVSN share their target similarity map, IVSN falls for distractors due to its greedy algorithm. **A)** The pattern on the wall fools IVSN. MM could not be computed on nnIBS's scanpath due to its short length. **B)** While looking for a bust, IVSN fixates on faces in paintings (fixation two, three and four).

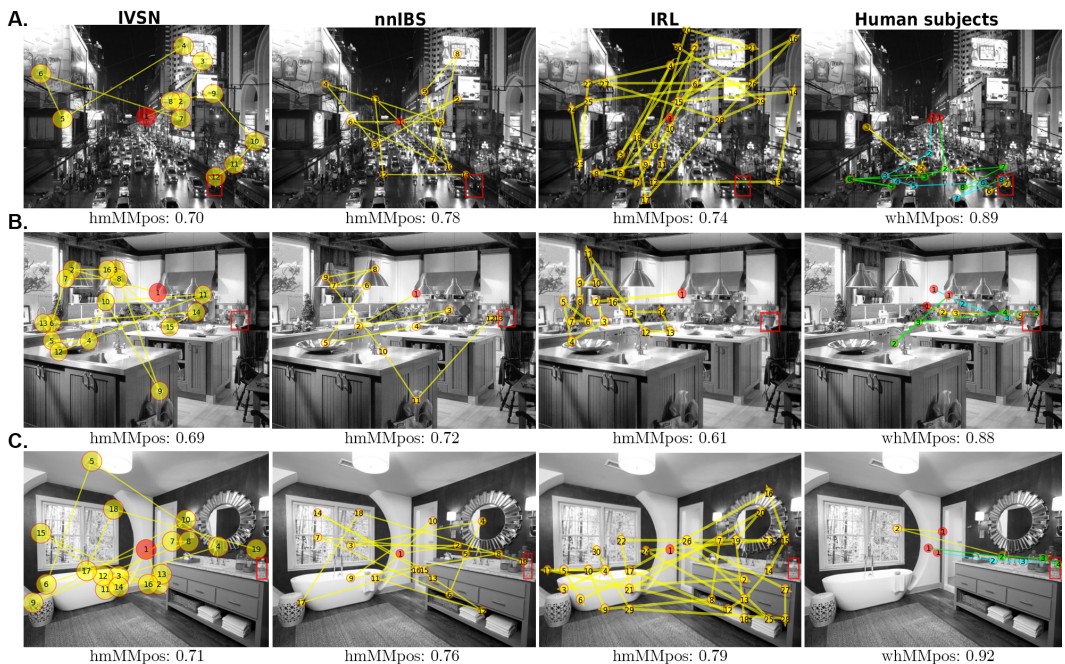

Figure S5: Search images where between-human Multi-Match was greatest and model-humans Multi-Match was lowest. **A)** Subjects immediately fixated on the road while looking for a car. **B)** The target is a potted plant, so humans only looked at the kitchen counter level. **C)** Similar to B), where the target is a bottle placed in a bathroom.

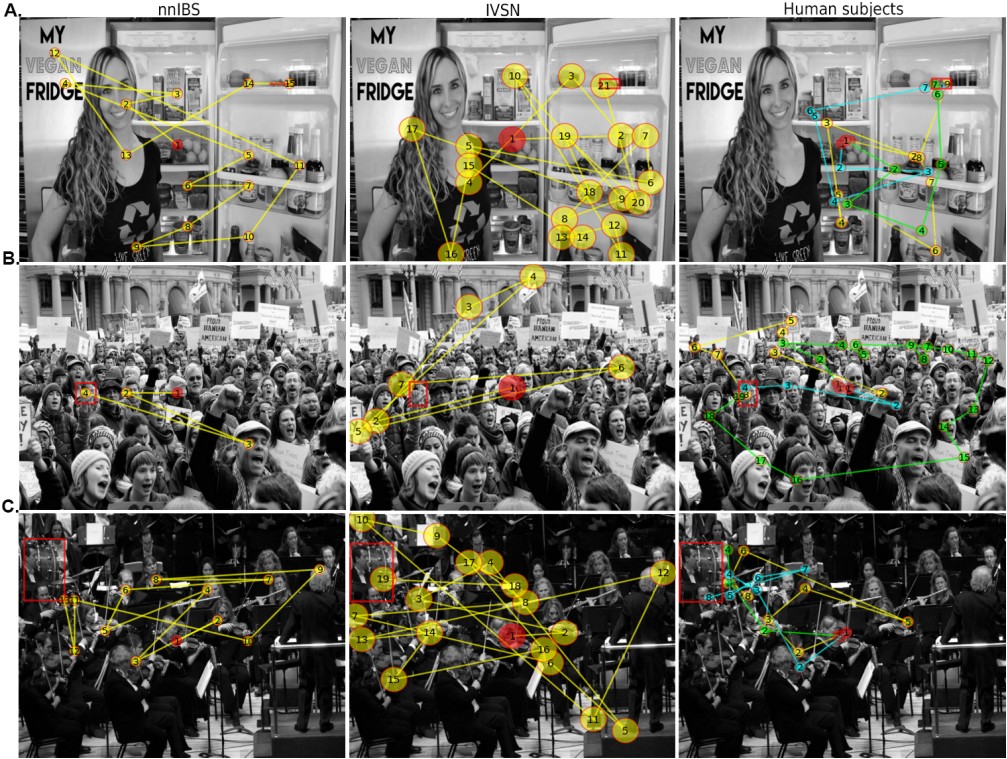

Figure S6: nnIBS, regardless of the target, fixates on human faces in the first few fixations. This pattern does not occur in human subjects.

## C  Data and Code Availability

All of the code used for this benchmark, including metrics, plots, and visual search models, alongside the preprocessed datasets, are available at the SVRHM branch in the `github.com/FerminT/VisualSearchBenchmark` repository.

