# OpenReview forum: "Benchmarking human visual search computational models in natural scenes: models comparison and reference datasets"
_NeurIPS.cc/2021/Workshop/SVRHM — SVRHM 2021 Poster_

### Official Review · Reviewer_QoKC · 2021-10-17
**useful benchmarking for visual search; accept**

**Rating:** 6
**Confidence:** 5

**Review:**

The paper summarizes existing visual search datasets in complex scenes and presents useful benchmarks on computational models in visual search. Besides, the authors performed incremental work on Ideal Baysian Seacher model and the results demonstrated marginal improvements over existing methods.

I agree with the authors that unifying benchmarks in visual search is necessary and beneficial for the entire community to move forward. The authors summarized a very nice set of benchmarking datasets, evaluation metrics and computational models despite several minor weaknesses (see the discussion below).

The paper is well-written. It is well-organized and easy to follow.

The source code is publicly available. I did not check the source code, but I believe the work is reproducible.

There are several suggestions for future work:

1. The current set of baselines is limited. There is a lack of competitive methods. For example, bottom-up saliency models (GBVS, DeepGaze2 and so on); and other top-down visual search models (previous visual search models from Zelinsky etal, https://you.stonybrook.edu/zelinsky/datasetscode/)

2. The authors introduced an incremental improvement of the original IBS model. However, based on the results in Fig 1 and 2, the improvements seem to be marginal, compared with IVSN. One advantage over IVSN is to generate alternative sequences of fixations. If so, have authors tried to look into the variability of visual scanpath introduced by IBS and humans? And how would this variability explain the consistency between humans and the model?

3. As the authors mentioned, the role of context is important in complex scenes of visual search. Adding prior contextual knowledge is useful in the future.

4. Minor: Line 20, "the mechanisms behind the visual processing of an image in the center of the visual field with no eye movements are well understood". Is there any evidence supporting this?

---

### Official Review · Reviewer_XdTu · 2021-10-24
**Paper provides thorough results and insights (in benchmark-form) for the task of visual search**

**Rating:** 8
**Confidence:** 4

**Review:**

This paper evaluates 3 different visual search models on 3 different visual search datasets (ones that were presented in the same papers as the models), taking great care to make the dataset preprocessing and evaluation of the models as comparable as possible. When it is possible to test multiple instantiations of a model’s design choices (attention map, similarity score, IoR patch size) the authors present results of these additional experiments (in the appendix) and include the best-performing variant in the main paper. This provides further modeling insights that can influence future work. Results comparing the 3 models on the 3 datasets are presented, demonstrating that different models fare well under different datasets/tasks (unsurprisingly: each model performs best on its own dataset), and some possible future research extensions are discussed to move progress forward in modeling human visual search.

Strengths of this work include:
+ a selection of one model out of each of the three main theoretical frameworks: CNN, reinforcement learning, Bayesian, to have a broad sample
+ a comparison of models and datasets and a discussion of how they differ
+ variations of the models evaluated to explore the influence of different design choices (e.g., CC/SSIM, attention map used, patch size for IoR) while keeping the best variants for comparison purposes in the main paper
+ evaluation was performed in a common computational framework, and the code will be made available to the research community
+ the conclusions offer good take-aways about model performances, strengths, and remaining gaps to human performance

What would help strengthen the paper further:
- the plots in 1D and 2E are not very informative, since it is hard to visually infer the correlation; the density of the points is not clear/color-coded from the plots, and the numerical values for the correlation (with significance stats) are not provided
- clarification on whether the different models naturally “stop looking”, whether they produce a fixed number of fixations across images, or whether the number of fixations/viewing time needs to be pre-specified in advance
- definitions of the evaluation metrics used; AUC is discussed in the paper, MM metrics refer to another paper (though for completion, it may be worth quickly defining them here - in the appendix - as well), but how is Corr measured?
- a discussion of some of the modeling/dataset trade-offs. The differences were listed, but as a reader, I was missing some commentary on the design choices (e.g., template-based versus categorical search task, data collection that stops when fixation lands on target versus when observer explicitly responds, using grayscale/color images, etc.) A lot of these discussions may be beyond the scope of the paper, but even a hint as to the trade-offs, e.g., in a table in the appendix, or references out to papers that perhaps discuss these tradeoffs might be insightful.

Overall, this is a very well-written paper with strong contributions. My slightly lower rating of confidence (and inability to select between a rating of 8 and 9) is based on the fact that I am generally familiar with the relevant literature, but not with the specific papers on visual search models, to be able to evaluate whether any key idea/model designs are missing, and whether the authors made proper treatment of the available models. The quality and thoroughness of the analyses inspires confidence, however. Another reason why this paper might be closer to an 8 rather than a 9 score is the generalizability of the paper topic (does it provide relevant insights to other researchers in CV who don't specifically work on human visual search tasks).

---

### Official Review · Reviewer_Q9Li · 2021-10-29
**Nice attempt at highlighting some of the problems and working towards comparative benchmarking**

**Rating:** 7
**Confidence:** 3

**Review:**

This paper discusses the problem of modelling targeted visual search tasks, and particularly looks at the problem of comparing how humans search against computational methods in terms of the gaze path. The paper highlights that existing work has often used different datasets and task formulations that make comparison hard, and makes a start in improving things by presenting an evaluation of a range of models within the same framework. In addition the authors propose several modifications to existing algorithms to see how they improve.

Overall I think this paper makes a nice contribution and provides a good foundation for further research, particularly in terms of models that more closely mimic human behaviour. Provision of the code used for the benchmark would be great for the community. Although I'd note that I'm far from an expert in this space, in terms of the overall discussion in the paper, would it be worth reflecting on earlier classical work (for example I'm aware the Itti worked on this ~15 or so years ago) and perhaps more broadly surveying current work (Ullman's work on course to fine search for example springs to mind).

---

### Decision · Program_Chairs · 2021-11-02

Accept (Poster)